# Extracellular Vesicles in Multiple Myeloma—Cracking the Code to a Better Understanding of the Disease

**DOI:** 10.3390/cancers14225575

**Published:** 2022-11-14

**Authors:** Justyna Iskrzak, Przemysław Zygmunciak, Irena Misiewicz-Krzemińska, Bartosz Puła

**Affiliations:** 1Medical University of Warsaw, 02-091 Warsaw, Poland; 2Institute of Hematology and Transfusion Medicine, Indira Gandhi Str. 14, 02-776 Warsaw, Poland; 3Department of Experimental Hematology, Institute of Hematology and Transfusion Medicine, Chocimska Str. 5, 00-791 Warsaw, Poland; 4Department of Hematology, Institute of Hematology and Transfusion Medicine, Indira Gandhi Str. 14, 02-776 Warsaw, Poland

**Keywords:** multiple myeloma, extracellular vesicles, diagnosis, prognosis

## Abstract

**Simple Summary:**

Multiple myeloma is a malignant hematological disease whose pathomechanism is not yet fully understood on the molecular level. Extracellular vesicles might be of significance in the evolution of this disease. Our aim is to present and systematize the latest findings concerning the role of extracellular vesicles in multiple myeloma, in order to enable an easier establishment of new research aims in this area. The results of such research could contribute to a better understanding of the disease and in consequence, result in the improvement of the diagnostic process of multiple myeloma and the invention of new therapies.

**Abstract:**

Multiple myeloma (MM) is a plasma cell-derived malignancy that stands for around 1.5% of newly discovered cancer cases. Despite constantly improving treatment methods, the disease is incurable with over 13,000 deaths in the US and over 30,000 in Europe. Recent studies suggest that extracellular vesicles (EVs) might play a significant role in the pathogenesis and evolution of MM. Further investigation of their role could prove to be beneficial in establishing new therapies and hence, improve the prognosis of MM patients. What is more, EVs might serve as novel markers in diagnosing and monitoring the disease. Great advancements concerning the position of EVs in the pathophysiology of MM have recently been shown in research and in this review, we would like to delve into the still expanding state of knowledge.

## 1. Introduction

Multiple myeloma (MM) is a hematological malignancy characterized by the accumulation of pathological monoclonal plasma cells, predominantly in the bone marrow [1]. Despite some advanced treatment options, the disease remains incurable with a 5-year relative survival of 57.9% (SEER 17 2012–2018) [2]. Current research into the underlying biology of MM has shed some light on the role of extracellular vesicles (EVs) in the pathogenesis and evolution of this condition [3]. MM evolves from the monoclonal gammopathy of undetermined significance (MGUS), in most cases via the acquisition of genomic mutations [4,5]. MGUS is defined by a serum non-IgM-type monoclonal protein of <3 g/dL (typically non-IgM-type), <10% clonal BM plasma cells, and the absence of end-organ damage [6]. MGUS may evolve to an intermediate stage—smoldering multiple myeloma (SMM), which is also asymptomatic, but with a higher risk of progression to malignant disease in the first 5 years than MGUS [7].

Extracellular vesicles are a heterogeneous group of particles that is steadily gaining acknowledgment in the medical field, as a vital part of the intercellular communication [8]. Not only do they mediate communication between healthy cells but are also an important part of pathological conditions, among which cancer is the most thoroughly studied [9]. Apart from malignancy, EVs’ functions are being investigated in many fields of medical research. Neurology is embracing EVs as a novel drug delivery system [10], a promising diagnostic tool distinguishing progressive supranuclear palsy and Parkinson’s disease [11], immunomodulators that may prevent inflammation-induced neural degeneration in ischemic patients [12], just to name a few. Cardiovascular research is focused on using EVs as a future acellular therapy for myocardial repair [13] and biomarkers of diverse clinical settings [14]. Studies on lung diseases have shown the possible application of circulating EVs as an exacerbation indicator in chronic obstructive pulmonary disease [15].

The classification and the nomenclature of EVs is an ongoing problem. The most commonly used classification is based on their biogenesis. Therefore, three groups are distinguished: exosomes are the EVs created by a fusion of multivesicular bodies (MVB) with the plasma membrane (PM); microvesicles (MVs) are established by direct budding from the PM; and apoptotic extracellular vesicles (ApoEVs) are the result of apoptosis (Table 1). This approach is not recommended by the International Society for Extracellular Vesicles (ISEV) since the consensus on the specific markers for each group is still to be reached [16]. Instead, general terms describing EVs’ physical characteristics, biochemical components, depiction of the condition, and/or the cell of origin, is recommended [16]. For instance, the classification, based on the size of EVs, which differentiates two classes: small EVs (diameter < 200 nm) and large EVs (diameter > 200 nm), is advocated [16].

As multiple studies fail to meet the criteria of inclusion in traditionally distinguished EV subgroups, to avoid any confusion, this article will apply the general term “extracellular vesicles” instead of using different subtype names, as proposed by the ISEV. In this review, the current state of knowledge on the role of EVs in the pathogenesis and progression of MM is presented.

## 2. Stages of MM Progression

Some genetic abnormalities (such as the acquisition of the hyperdiploidy, and the translocations involving the immunoglobulin heavy chain gene locus) are the trigger for the induction of plasma cell malignancy and can be detected in the MGUS and the SMM, years before proper MM development [1]. It seems that the main genetic mutations in the progression of myeloma take place in the early stage of the disease between MGUS and SMM, while the meaning of mutations between SMM and MM is more elusive. There exist two models of progression from SMM to MM. The first one, called “static progression”, is characterized by the same subclonal architecture of MM and SMM, suggesting that some factors other than genetic factors could play a role in the passage between these two stages. The second model, “spontaneous evolution”, depends on the random acquisition of new mutations, which also leaves space for further investigation [23]. It is proposed that the communication between the MM and the microenvironment of the bone marrow (BM), notably by EVs, is an important factor in the MM evolution [3]. Cell proliferation and tumor growth, angiogenesis, matrix remodeling and metastasis, immunosuppression, and drug resistance are some of the processes mediated by EVs and this article is an elaboration on them. It is known that a single EV can exert influence in multiple pathways of MM progression. However, simplification will be used for the sake of the clear structure of this review.

### 2.1. Tumor Growth and Proliferation

The microenvironment surrounding a tumor plays a vital role in its progression. As the tumor advances, it develops a network of connections between various BM populations to induce the proliferation of its cells, assuring its growth. Especially, the cross-talk with bone marrow mesenchymal stromal cells (BM-MSCs) is of great importance in this process [24] (Figure 1).

Tumor-growth-inducing effects of EVs, deriving from multiple myeloma mesenchymal stromal cells (MM-MSC), are mediated by their modified cargo. The amended content of the EVs includes increased pro-proliferative protein levels (namely IL-6, CCL2, junction plakoglobin, and fibronectin) and an altered nucleic acid load [24] (Figure 1). Decreased levels of microRNA (miR)-15a and augmented levels of long-non-coding RNAs (lncRNAs)—LINC00461 were found in proliferation-inducing MSC-derived EVs [24,25]. miR-15a, as well as miR-16, acts as a regulator of the BCL-2 expression that is a well-known apoptosis suppressor [26], whereas LINC00461 has a down-regulating ability on miR-15a and miR-16 by direct binding [25]. Therefore, it appears that the mechanism of the direct binding of miR-15a and miR-16 by LINC00461, might be the cause of decreased miR-15a levels found in the Roccaro et al.’s study and consequently, the increase of proliferation of malignant cells.

What is more, MSCs possess the ability to transform into cancer-associated fibroblasts (CAFs), in a process mediated by MM-derived EVs [27,28] (Figure 1). These EVs had an augmented content of miR-21 and miR-146a, which promotes the proliferation and transformation of MSCs into CAFs, by inducing the expression of IL-6, stromal-derived factor 1 (SDF-1), fibroblast-activated protein (FAP), and α-smooth muscle actin (α-SMA) genes [27] (Figure 1). Moreover, miR-146a transferred from the MM cells to MSCs increases the expression of the CXC ligand 1 (CXCL1), interferon-γ–inducible protein 10 (IP-10), and the CC chemokine ligand 5 (CCL5) in MSC [28] (Figure 1). This process is mediated by the Notch pathway and its inhibition leads to the down-regulated cytokines production, therefore decreasing the MM cell growth [27].

As aforementioned, miR15a and miR-16 are MM suppressing factors, but interestingly other miRNAs, notably miR-10a, deriving from bone marrow stromal cells (BMSCs) and released in EVs, works in an opposite manner, encouraging the MM cell proliferation and tumor growth [29]. Probable gene targets in the MM cells include β-transducin repeat-containing E3 ubiquitin protein ligase (β-TRC), whose expression was augmented by the miR-10a overexpression [29] (Figure 1). β-TRC is a part of the Skp1, cullin, F-box proteins (SCF) E3 ligase complex, which mediates the degradation of the Rap1 GTPase-activating protein 1 (Rap1GAP) required to induce the cell proliferation [30]. In line with this mechanism, the inhibition of β-TRC suppressed a pro-proliferative effect of miR-10a [29]. What is more, the bioinformatic analyses predicted ephrin receptor A8 kinase (EphA8) as a possible target gene for miR-10a, creating a novel potential miR-10a-EphA8-SEMA5A pathway [31]. Nevertheless, extensive research, including laboratory experiments, is yet to be performed to assess the value of this prediction. Interestingly, miR-10a induces apoptosis in BMSCs when the EV formation is blocked. Thus, the inhibition of the EV release might be a novel treatment for MM [29].

Stem cell-derived EVs increase the MAPK activation within the MM cells, nevertheless, the proliferation induction is dependent on the cellular origin of the cell [32]. Within 24 h, the decrease in the activation of MAPK was seen in the normal donor-deriving EV group, whereas myeloma EVs initiated the continuous MAPK activation, which translated into an augmented viability, proliferation, and translation initiation [32] (Figure 1). Interestingly, similar findings regarding the antiproliferative effect of EVs deriving from normal donors cells, were found elsewhere [24]. Hence, a thorough analysis of EVs’ content across the BM populations could be beneficial to identify possible common antiproliferative pathways which might be used as new therapeutic targets.

All things considered, communication between MM and diverse cellular compartments of BM is of great importance, considering the induction of the proliferation and tumor growth. Nevertheless, MM cells may also induce this process in an auto-stimulatory way via the internalization of the CD147-positive EVs [33] (Figure 1). EVs derived from human MM cell lines of the decreased CD147 expression, have been shown to induce the proliferation of MM cells in a weakened way [33]. The research investigating the EVs’ ability to serve as a carrier in the auto-stimulation processes of malignant cells may be a novel and important point of view, regarding EVs’ place in the tumor biology.

### 2.2. Progression

One of the main characteristics of cancer that greatly complicates the course of the disease is its ability to disseminate. To achieve its progression, MM requires the cooperation of its cells with the bone marrow of different skeletal sites [34]. Needless to say, efficient communication between these two must be established. There exists a number of studies that demonstrate the interaction between BM-MSCs and MM cells, which in turn, results in the tumor cell migration, proliferation, creation of a convenient microenvironment (for instance, through angiogenesis or osteoclastogenesis), and finally metastasis [35,36,37]. Recent findings identify EVs as an important means of communication between these two [24,32].

Throughout the disease, MM cells manage to influence the BM-MSCs, so that they start cooperating with them in order to enable metastasis [32]. It is not a peculiar characteristic of multiple myeloma, as it has been also shown in research, concerning other cancers [2,38,39,40].

The MM-EVs influence the BM-MSCs, and through this, the preconditioning of macrometastasis is possible. MM-EVs have been shown to influence BM-MSCs through the signal transducer and activator of transcription 3 (STAT3) and c-Jun N-terminal kinase (JNK) phosphorylation, in order to increase their survival [41]. As a result of the preconditioning, BM-MSCs can produce altered EVs (MM MSC-EVs) [41].

One of the components of great importance in the MM progression, are miRNAs derived from MM-EVs. miR-146 and miR-21, originating from MM-EVs, affect the cytokine production in BM-MSCs and favor their differentiation into CAFs. This is achieved through the activation of the cytokines cascade, in which molecules, such as IL-6, IL-8, CXCL1, and CCL5, to name a few, are secreted [42,43]. As a result, they positively affect the growth and migration of tumor cells. Moreover, the BM-MSCs secrete EVs containing miR-10a, which is subsequently incorporated by the MM cells, promote their proliferation and help create a suitable microenvironment for the MM cells to survive and, as already has been described above [29].

Not only cargo of the MM-EVs, but also their surface molecules, have been shown to exert a strong influence on the BM of MM patients. CD138 (also known as Syndecan-1) is a cell surface proteoglycan containing heparan sulfate, whose presence on the peripheral EVs was found to be correlated with the presence of bone lesions in MM patients [44]. The suggested mechanism which could explain this pathway is linked to the role of fibronectin. This molecule is bound to CD138 and exerts an influence on the MM cells via the p38 and pERK pathways, which in turn, leads to the expression of Dickkopf-related protein 1 (DKK1) and metalloproteinase-9 (MMP-9), two molecules with a favorable effect on the MM progression [45].

In MM patients, the extracellular circulating 20S proteasomes are also detected and their source is elusive. However, EVs are a highly probable source of them, as seen in the study focusing on T lymphocytes microvesicles [46]. The proteasome system’s function is the degradation of proteins involved in cellular processes, such as apoptosis, transcription, antigen processing, and hence, in the proliferation of cells, both normal and malignant [47]. The importance of 20S proteasomes in MM is highlighted by the fact that the circulating proteasome level is significantly elevated in active, versus smoldering MM, suggesting its role in the MM progression [48].

### 2.3. Angiogenesis

It has been shown in various studies that EVs also play a key role in the induction of angiogenesis [41], one of the ways in which cancer gains its abilities for further survival [49].

A study analyzing the naive MM-EVs and EVs pretreated with bortezomib (Bi-EVs), has shown that MM-EVs contain pro-angiogenic factors and internalized by EC, enhance the migration and proliferation of the latter. Bortezomib altered the levels of the pro-angiogenic and inflammatory proteins in MM-EVs. The expression of VEGF-R2 and several growth factors (such as VEGFA, platelet-derived growth factor-BB (PDGF-BB), angiogenin, and urokinase plasminogen activator receptor (uPAR)) were significantly reduced on Bi-EVs, compared to naive MM-EVs. The exposure of endothelial cells to naive MM-EVs and Bi-EVs was analyzed and apparently, exposure to naive EVs induced a trend of an increase in endothelial cell (EC) proliferation and migration, influencing the MAPK cascade mechanism, whereas Bi-EVs inhibited the EC proliferation and migration. The study demonstrated the cross-talk of EVs and EC, which results in angiogenesis in the progression of MM [50] (Figure 2).

The formation of new vessels is known to be promoted by hypoxia, which is normally induced by the aggressive proliferation of MM cells in BM [51,52]. Under these conditions, various substances are secreted by the cancer cells to facilitate the formation of new vessels in an otherwise hostile environment [51]. The hypoxia-resistant MM cells (HR-MM cells) produce more EVs than their parental cells. The RNA components of these EVs seem to be of great importance as they differ significantly from those of the EVs of the parental cells. Some microRNA (notably miR-135b) of the HR-MM EVs play a role in blocking factor-inhibiting hypoxia-inducible factor 1 (FIH-1) in endothelial cells, which in turn promotes the endothelial tube formation [47]. Similarly, piRNA is an important regulator in the evolution of MM. piRNA-823 plays a significant role in angiogenesis, inducing the tumor growth and proliferation [53]. piRNA-823 increases in vitro the level of VEGF and IL-6, as well as restricts the apoptosis of the infected endothelial cells through the inhibition of the caspase-3 activation, downregulation of the BAX expression, and the production of ROS and NO [53] (Figure 2).

RNAs are not only EV-derived inductors of angiogenesis. Several cytokines and other proteins known to be involved in the formation of vessels, such as the vascular endothelial growth factor-1 (VEGF), angiogenin, and CD147, to name a few, were detected as a cargo of EVs [39,54,55]. Worth mentioning is the effect of CD138 on the endothelial cells whose gene expression correlated with the increased density of microvessels in the BM in patients with MM and MGUS [56] (Figure 2). It is yet to be researched if CD138 influences angiogenesis at the premetastatic niche.

Several studies found that not only MM-EVs can contribute to angiogenesis in the evolution of MM. Endothelial cells also generate EVs, which influence tumor invasion and angiogenesis. They release EVs containing metalloproteinases (MMPs) that lead to proteolysis, necessary for metastasis and free angiogenesis [57]. miR-126 is present in EC-derived EVs and exerts pro-angiogenic actions on the microenvironment and other ECs [58] (Figure 2). Moreover, miR-126 contributes to establishing a pro-angiogenic microenvironment and promotes the hindlimb ischemia repair [59]. It is thought to act through the VEGF/PI3K/Akt pathway [60].

### 2.4. Matrix Remodeling and Osteolysis Induction

The formation of a suitable niche and dissemination are necessary steps in the neoplasm progression [61]. One of the roles of EVs in the MM evolution appears to be matrix remodeling and specifically, the induction of osteolysis, which in turn, facilitates the dissemination of MM and creates non-hostile conditions for the tumor to progress (Figure 3). The BM provides a highly convenient niche for multiple myeloma and facilitates the further dissemination of the MM cells [24]. Therefore, it is not surprising that MM cells tend to diffuse to distant skeletal sites [62]. This affinity to the bone marrow results in what we call multiple myeloma-associated bone disease (MBD), which significantly reduces the quality of life of patients and increases morbidity [63].

A key function of EVs is the induction of bone resorption, whose role varies greatly from “freeing up” the space for the MM cells to reside to secreting factors influencing the tumor proliferation, such as TGF-β [64]. Physiologically, there exists a balance between the osteoclasts (OCs) and osteoblasts (OBs). Nevertheless, during the MM evolution, this balance is disturbed by the advantage of OCs which leads to osteolysis [44]. The multiple components of EVs take part in different pathways, in order to establish a suitable niche for the MM cells to disseminate. A study demonstrating the effects of MM EVs on the imbalance between OCs and OBs and hence, osteolysis has been conducted on the 5TGM1 MM murine model. The 5TGM1 EVs were shown to enhance the OCs activity as well as block the OBs differentiation and impede their functions, as a result of the transfer of DKK-1, the Wnt signaling inhibitor, which in turn, induced the reduction of the Runt-related transcription factor 2 (Runx2), osterix and collagen1A1 in OBs [65].

Another cause of the disturbed balance between OCs and OBs is a protein component of MM-EVs playing a role in osteoclastogenesis-amphiregulin (AREG), which is a ligand of the epithelial growth factor receptor (EGFR) [66]. AREG presents a vast range of functions. It activates EGFR in the osteoclast precursors, leading to their differentiation into mature OCs. It also promotes the secretion of osteoclastogenic cytokines, such as IL-8 from BM-MSCs. Finally, AREG blocks the mesenchymal stem cell differentiation into the OBs [67]. The study also shows another pathway in which the MM-EVs promote osteolysis: they have the ability to increase the secretion of RANKL and decrease the release of osteoprotegerin. The MM-EVs play an important role in inducing osteolysis, as they contribute greatly to the migration of the osteoclasts’ precursors and their survival. This is achieved by increasing the CXCR4 (CXC chemokine receptor type 4) and by upregulating the expression of Bcl-2 and AKT phosphorylation. What is more, MM-EVs promote bone resorption by augmenting the expression of important osteoclastogenic enzymes (TRAP, CTSK, MMP9) and inhibit the apoptosis of the osteoclast precursors by blocking the caspase-3 activity [68].

The RNA content of MM-EVs has a vital role in osteolysis in MM [69]. According to recent studies, MM-EVs contain miR-135b, which by being incorporated by BM-MSCs, targets and downregulates SMAD5 [47]. SMAD5 is a protein that, together with bone morphogenetic proteins (BMPs), govern the process of osteogenesis [47]. Its downregulation disturbs osteogenesis [47]. Another microRNA, miR-21, produced through the STAT3 pathway and stored in MM-EVs, was shown to be involved in promoting the osteoclast differentiation [70,71]. It was shown that miR-21 targets the 3′UTR of osteoprotegerin (OPG) mRNA, which in turn, contributes to the imbalance of RANKL/OPG—key players in the physiologic bone remodeling [71]. Not only miRNAs, but also lncRNAs have been identified in studies as significant in the development of MBD. The transfer of lncRUNX2-AS1 from the MM-EVs to marrow stem cells can inhibit the RUNX2—an essential transcription factor regulating the osteoblast differentiation, which leads to the inhibition of osteogenesis through the interference in the splicing of RUNX2 [72].

The MM-EVs facilitate the MM cell dissemination also, thanks to their procoagulant content (tissue factor [TF] and procoagulant phospholipids). The components of the EVs lead to the start of the cascade of coagulation, in which platelet activation plays a vital role [73]. The platelets have been already shown to exert an effect on the metastatic niche through the expression of such cytokines as TGF-β [74]. Another important platelet-derived cytokine in the MM progression is IL-1β, which has been shown to cause an increased tumor cell engraftment in vivo [75].

Interestingly, the chemotherapeutics that are used for MM treatment (such as bortezomib, carfilzomib, and melphalan) may increase the osteoclastogenic function of MM-EVs. These drugs increase the level of heparanase in MM-EVs, which in turn activates the cell surface protein, syndecan-1, which interacts with BM-MSCs to promote tumor dissemination. It also activates the ERK pathway which leads to the secretion of osteolytic factors, MMP-9 and RANKL [76]. Whether this mechanism has a significant clinical impact on the treatment outcome of MM patients remains to be clarified.

### 2.5. Immunosuppression

The hematopoietic compartment of a BM consists of myeloid cells, T and B lymphocytes, natural killer (NK) cells, and osteoclasts and it is an important factor in an immune response toward malignancy [77]. A cross-talk between BM cells serves firstly as a killer and then as an inhibitor of malignant development. Interestingly, with tumor progression, it appears that some BM cells cooperate with MM, promoting the immune escape [41,78]. This corresponds with the theory of Schreiber et al., stating that there are three main phases of cancer immunoediting: elimination, equilibrium, and escape [79]. In the first phase, both innate and adaptive immune systems effectively detect and kill the cancerous cells, but some tumor cell variants might survive it and enter a second phase—equilibrium, where the growth of the tumor is inhibited by the immune system. This phase was observed to be the longest one and some researchers suggest that the benign conditions preceding MM, thus MGUS and SMM, could match with equilibrium [80].

The immune escape of the tumor might be executed by the transformation of the malignant cell or the immune system of the host. Alterations in the hematopoietic populations were described extensively as possible mechanisms of the immune evasion, both from the innate and adaptive immunities [81,82,83,84,85,86]. As mentioned before, microenvironmental factors, such as EVs, could play an essential role in the process.

Indeed, EVs exert a direct effect on the myeloid-derived suppressor cells (MDSCs), increasing their viability and proliferation by up-regulating the STAT3 pathway and increasing the pro-survival proteins B-cell lymphoma-extra-large (Bcl-xL) and myeloid cell leukemia-1 (Mcl-1) [41]. Moreover, the cargo of MM-EVs includes PGE2 and TGFβ, which stimulate the neoangiogenesis and MDSCs buildup in the tumor site [87]. When stimulated by the MM-derived EVs, the transition to the highly immunosuppressive subpopulation CD11b+Ly6GlowLy6C+, is induced [41]. This subclass mediates the immune evasion by overexpressing the inducible nitric oxide synthase (iNOS), which promotes the T-cell inhibition via the induction of hydrogen peroxide and peroxynitrite. These compounds work in various mechanisms, collectively blocking the T cell receptor (TCR) dependent activation of T lymphocytes [88]. What is more, MM cells, stimulated by EVs, produce an augmented amount of inflammatory cytokines, such as IL6 and VEGF, which further aggravate the MDSCs accumulation, tumor growth, and other immunosuppressive mechanisms, hence inducing a favorable environment for malignancy [87,89].

What is more, MM cells with chromosome 13 deletion (Del13) secrete increased amounts of EVs within which reduced miR-16 levels were detected. When stimulating monocytes, these EVs induced the differentiation to the protumoral M2 macrophage population. miR-16 is a negative modulator of the NF-kB pathway, which is involved in the M2 polarization by inducing the expression of M2-specific cytokines, including IL-10, IL-8, and TNF-α [90].

Moreover, EVs can also serve as anti-tumor immune response inducers and boosters. As tumor antigen transporters (namely mucin 1 (MUC1), survivin) they can activate the T-cell response mediated by dendritic cells (DC) [91]. In addition, EVs are well-documented positive modulators of the NK cell response. NK cells might be stimulated by some MM-derived EVs expressing Hsp-70 and interleukin 15 receptor α (IL-15Rα) [92,93,94]. Similarly, DC-derived EVs expressing IL-15Rα might trigger the NK cell response [95]. These immunity-inducing properties of the EVs led to the ongoing research of antitumor EV-based vaccination, which hopefully will be established soon [96].

### 2.6. Drug Resistance

Despite the introduction of new therapeutic agents, drug resistance remains one of the biggest problems in MM patients and the median overall survival in triple-class refractory patients is only 5.6 months [97]. Drug resistance in MM is connected to both adjustments in MM cells (i.e., the upregulation of Pgp [98], the mutation in the proteasomal system [95], changes in CD expression [99]) and the microenvironmental adaptations (environment-mediated drug resistance (EMDR)) where the EV mediated cross-talk is considered to play an important role [100,101].

In previous chapters, we described some of the adaptations that increase MM survival. Indeed, the aforementioned mechanisms play a vital role also in the drug resistance induction, therefore the endurance of the cancer cells. A cross-talk between the endothelial and MM cells leads to the creation of the vascular niche, whereby partial protection from anti-tumor agents is provided [102]. Furthermore, an interaction between osteoblasts, osteoclasts, and stromal cells prevents the pro-apoptotic actions of the anticancer drugs [100]. Moreover, the MM-induced immunosuppression impairs the T-cell activity, avoiding the antibody-dependent cell-mediated cytotoxicity (ADCC) [41].

There are three main groups of anti-myeloma drugs: proteasome inhibitors (i.e., bortezomib, carfilzomib, ixazomib), immunomodulators (i.e., thalidomide, lenalidomide, and pomalidomide), and monoclonal antibodies (daratumumab, elotuzumab, isatuximab). Interestingly, the EV-mediated resistance was described for each group, which shows that EVs play an important role in MM survival (Figure 4).

#### 2.6.1. Proteasome Inhibitors (PI)

Proteasome inhibitors are commonly administered as the first-line treatment in MM patients, as well as in refractory and relapse patients [103]. These agents bind to the catalytic site of the 26S proteasome, therefore, suppressing its degrading activity and deregulating the ubiquitin-proteasome pathway (UPP) [104]. This alteration leads to the cellular toxicity in direct (apoptosis induction via JNK and p53 pathways) and indirect mechanisms (reduced degradation of pro-apoptotic molecules) [105].

It is known that soluble factors derived from BMSC, encourage resistance toward a drug-induced apoptosis [106]. The mechanism behind this process is linked to the inhibition of caspase-3, caspase-9, and PARP cleavage and therefore the increased viability of the malignant cells in the presence of bortezomib. Interestingly, not only MM BMSC-derived EVs, but also EVs isolated from healthy donors improve the viability of MM cells [107] (Figure 4).

Moreover, BMSC is not a single population mediating an acquisition of the drug resistance against PI. Stem cell-derived EVs contain lncRNA PSMA3-AS1, which is involved in the positive regulation of the α7 proteasome subunit PSMA3 in MM cells (Figure 4). An augmented expression of both PSMA3 and PSMA3-AS1 mRNA corresponds with the PI resistance, overall survival (OS) and progression-free survival (PFS). Thus, these EVs are being examined as a potential biomarker of MM [108].

#### 2.6.2. Immunomodulators

The EVs’ ability to transfer drug resistance was also described, regarding immunomodulators. The lenalidomide-sensitive MM cells were able to acquire resistance when cocultured with resistant MM cells. Interestingly, the acquired resistance corresponded with a stronger adherence of the sensitive cell lines. Lenalidomide resistance was correlated with the overexpression of SORT1 and LAMP2 genes that are responsible for a higher EV release (Figure 4). Sadly, no cargo of the EVs was described in the study and the researchers weren’t able to name a specific mechanism binding all of the aforementioned findings together, therefore the topic needs further clarification [109].

#### 2.6.3. Monoclonal Antibodies

The introduction of monoclonal antibodies in the MM treatment is a great success in the present hematology, improving the overall survival and progression-free survival [110]. Daratumumab and isatuximab target CD38 while elotuzumab targets SLAMF7, the proteins that are overexpressed on the surface of MM cells [111,112]. However, we need to remember that the content of the EVs corresponds with the parental cell. Indeed, as mentioned before, the vesicles derived from the myeloma cells overexpress CD38, just as the cell of origin [113]. A study on HER-2-positive breast cancer cells indicated that HER-2-positive EVs can bind to trastuzumab, therefore decreasing the bioavailability of the drug [107]. In conclusion, it seems possible that MM-derived EVs that express CD38 can have a similar effect on both daratumumab and isatuximab (Figure 4). Less is known about the SLAM family as an EVs cargo, therefore this topic needs further evaluation.

## 3. Conclusions

As demonstrated in this article, MM is highly dependent on other cells and the cooperation between them is essential for the tumor progression. A cross-talk between cancer and its microenvironment is partially mediated by EVs, which explains a growing interest in this matter worldwide. EVs are abundant in different types of cargo. They can contain molecules, such as proteins, cytokines, chemokines, miRNA, lncRNA, and more [39,47,53,54]. A further elaboration on the precise relationship between cancer and EVs is necessary as it might lead to the invention of new cancer therapeutic strategies in the future. The knowledge about the EVs’ involvement in the pathogenesis of MM might add up significantly to the already ongoing works concerning a dendritic cell-based vaccination in MM [91,114], as it has been demonstrated in the case of the tumor-derived EVs’ influence on immunointerventions against melanoma [115]. Another possible therapeutic approach could repose on the role of apoEVs, derived from MSCs. They have been shown to induce the Fas-mediated apoptosis of MM cells in a mouse model, thus, prolonging the life span of MM mice [116].

An increasing number of studies concerning EVs in the pathogenesis of multiple myeloma is an indicator of their importance in modern research. Indeed, some unexpected results were published concerning EVs, which 30 years ago were considered cellular debris. Various studies bring more and more clarity, as far as the structure and cargo of EVs are concerned. Despite the statement from the International Society for Extracellular Vesicles and the updated guidelines, the approach toward the research on EVs is not unified and the nomenclature and study designs lack the needed consistency. Perhaps, the guidelines that will be published this year might prove to be of significance in this matter. The improvement of the tools for better EV differentiation and collecting is also needed. Better methods will allow more precise analysis of such a vast topic as EVs.

EVs have been detected in different body fluids, such as blood plasma [117], saliva [118], bronchoalveolar fluid [119], cerebrospinal fluid [120], to name a few. EVs contain different molecules, which could reflect the pathogenesis and progression of a disease, even at an early stage [121,122]. What is more, EVs can be detected using minimally invasive procedures, which is advantageous for multiple reasons, the possible shortening of the time of the diagnostic pathway, a faster implementation of therapy, the possibility of patients, who otherwise would be unfit for certain invasive diagnostic procedures to obtain a thorough diagnosis, and lastly, the economic reasons, including the lowered costs of diagnosis and follow-up. The use of modern techniques, such as phage display technology, in the identification of EVs, which has recently been demonstrated on an animal model, might lead to a less invasive and more personalized diagnostic process [123]. Thanks to these characteristics, EVs could be employed in everyday practice as a perfect tool for early detection as well as a marker of the progression of a disease. As far as the research in MM EVs is concerned, a different EV cargo expression and a difference in the quantity of EVs between MGUS, SMM, MM, and healthy donors, have been shown [124,125]. What is more, a recent study identified certain miRNA from circulating EVs (namely miR-18a and Let-7 microRNA precursor) as a prognostic marker of MM [126]. These findings suggest that circulating EVs could constitute a novel marker of the MM progression and possibly a diagnostic marker in a more distant future. This, in turn, would surely diminish the effects of a bone marrow biopsy, an invasive procedure that can be aggravating, especially for the elderly people that count as the biggest myeloma patients’ group. A marker that could not only distinguish MM from other conditions but also demonstrate the stage of the disease would positively influence the quality of life of multiple myeloma patients.

To summarize, the article presents the rising importance of EVs in the understanding of MM pathogenesis. Hopefully, when better understood, EVs might serve as a useful tool in the diagnosis and treatment of MM patients.

## Figures and Tables

**Figure 1 cancers-14-05575-f001:**
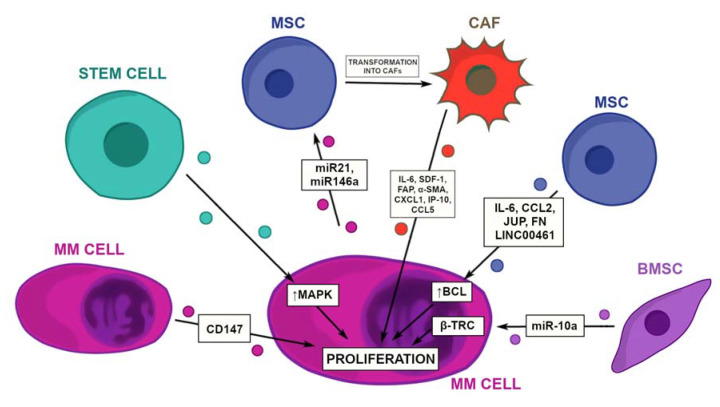
EVs’ involvement in the induction of the proliferation of MM cells. Interactions between numerous stromal cells lead to tumor growth. MSCs send pro-proliferative cytokines, which increase the BCL-2 content inside MM cells, inducing its viability and survival. When stimulated by MM, EVs that contain miR-21 and miR-146a, transform into CAFs, which induce the proliferation of tumor cells by multiple cytokines. Stem cell-derived EVs induce MAPK, which by stimulating multiple kinase cascades, regulates many pathways, including the proliferation-inducing routes. BMSC release miR-10a inside their EVs that induce β-TRC within the MM cells. β-TRC is necessary for activating the cell cycle-regulating proteins, such as cyclins and transcription regulators, thus promoting the proliferation. Interestingly, MM can also stimulate its own growth by releasing CD147-positive EVs. α-SMA—α-smooth muscle actin, BCL-2—B-cell lymphoma 2, BMSC—bone marrow stromal cell, β-TRC—β-transducin, CAF—cancer associated fibroblast, CCL—CC chemokine ligand, CD—cluster of differentiation, CXCL—CXC ligand, FAP—fibroblast-activated protein, IL—interleukin, IP-10—Interferon-γ–Inducible Protein 10, JUP—junction plakoglobin, LINC00461—long intergenic non-protein coding RNA 461, MAPK—mitogen-activated protein kinase, miR—microRNA, MM—multiple myeloma, MSC—mesenchymal stromal cell, SDF-1—stromal-derived factor 1.

**Figure 2 cancers-14-05575-f002:**
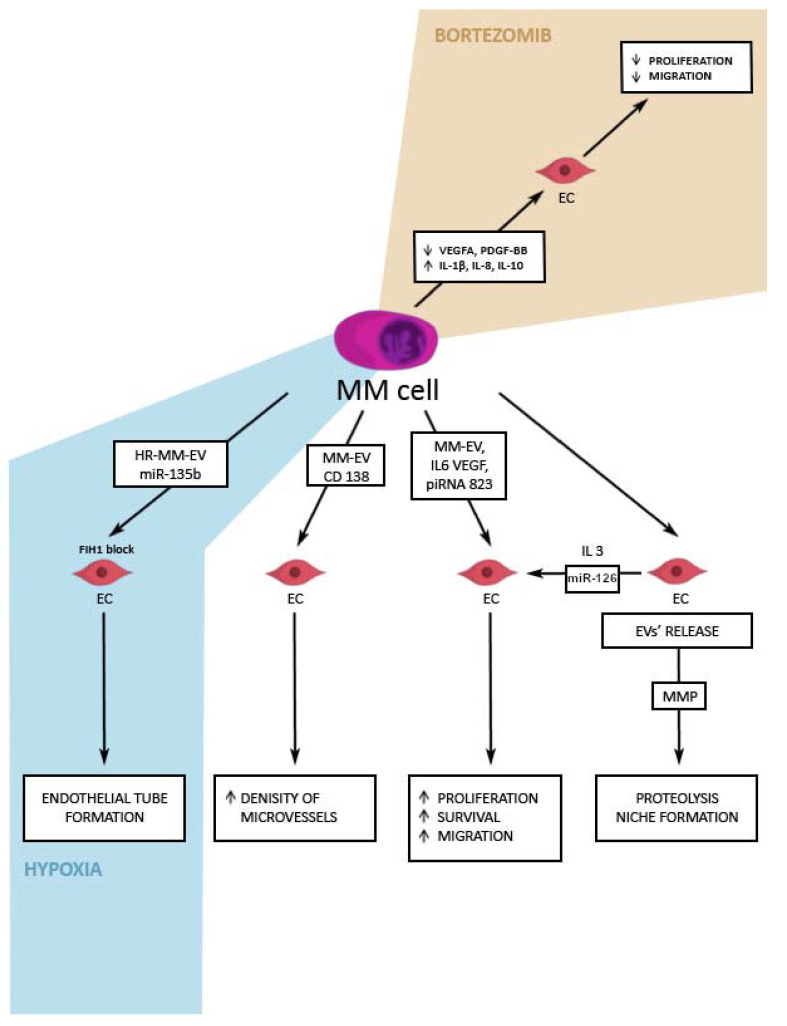
EVs’ involvement in the process of angiogenesis. MM-EVs interact with ECs and promote angiogenesis and hence, the further survival of tumor cells. MM-EVs’ component, piRNA-823, incorporated by the endothelial cells, could increase the level of VEGF and IL-6 and therefore, lead to the increased proliferation, survival, and migration of endothelial cells. A surface molecule of MM-EVs, CD138, could interact with ECs—its gene expression is correlated with the density of the microvessels in the stages of MM and MGUS. Different conditions alter the composition of MM-EVs. MM-EVs exposed to hypoxia contain miR-135b, which blocks FIH1 in the endothelial cells and hence, leads to the endothelial tube formation (blue background in the figure). MM-EVs pretreated with bortezomib (proteasome inhibitor used in the treatment of multiple myeloma) (beige background in the figure), contained lower amounts of the pro-angiogenic factors (VEGFA, PDGF-BB, uPAR) and higher amounts of inflammatory interleukins (IL-1β, IL-8, IL-10), than the MM-EVs derived from the naive cells. What is more, the exposure of endothelial cells to these pretreated with bortezomib EVs, inhibited the EC proliferation and migration. d) EC-EVs also play a role in the angiogenesis promotion—they produce MMPs which lead to proteolysis and niche formation. Another component of EC-EVs, miR-126 has been shown to exert a pro-angiogenic influence, concordant with the IL-3 action on the microenvironment and other ECs. ECs—endothelial cells, EC-EVs—endothelial cell-derived extracellular vesicles, FIH1—factor-inhibiting hypoxia-inducible factor 1, IL-1β—interleukin 1β, IL-3—interleukin 3, IL-6—interleukin 6, IL-8—interleukin 8, IL-10—interleukin 10, MGUS—monoclonal gammopathy of undetermined significance, miR-126—microRNA-126, MM—multiple myeloma, MM-EVs—multiple myeloma-derived extracellular vesicles, MMPs—metalloproteinases, PDGF-BB—platelet-derived growth factor-BB, piRNA-823—PIWI-interacting-RNA 823, uPAR—urokinase plasminogen activator receptor, VEGF—vascular endothelial growth factor VEGFA—vascular endothelial growth factor A.

**Figure 3 cancers-14-05575-f003:**
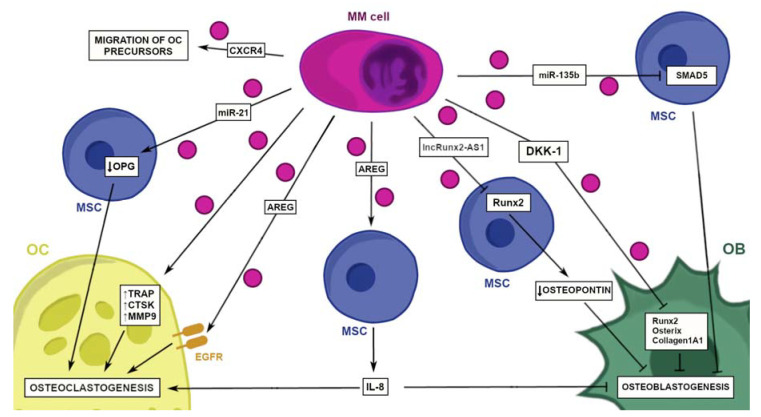
The influence of MM-EVs on osteolysis and bone formation. Induction of osteolysis appears to be one of the roles of MM-EVs. MM-EVs have been shown to reinforce the advantage of OCs in the imbalance of OCs and OBs. When incorporated by BM-MSCs, a component of MM-EVs, miR-135b, downregulates SMAD5, a protein which, together with BMPs, is responsible for the process of osteogenesis. The downregulation of SMAD5 disturbs the process of osteogenesis. DKK-1, an inhibitor of the Wnt signaling pathway is another component of MM-EVs. When incorporated by the osteoblast precursors, DKK-1 blocks the Wnt pathway and leads to the diminished expression of Runx2, osterix and collagen1A1, resulting in the disturbance of osteogenesis. lncRunx2-AS1 integrated by BM-MSCs targets the gene Runx2 and downregulates it. This results in the downregulation of the osteopontin expression and eventually leads to the inhibition of osteogenesis. Another important component of MM-EVs is AREG, a ligand of EGFR. It activates EGFR in the osteoclast precursors, leading to their differentiation into mature OCs. It also promotes the secretion of osteoclastogenic cytokines, such as IL-8 from BM-MSCs. Another pathway in which MM-EVs promote osteolysis is linked to the balance between RANKL and OPG. miR-21, found in MM-EVs, targets the OPG gene in BM-MSCs and lowers its expression. This leads to the imbalance of RANKL/OPG and leads to osteoclastogenesis. MM-EVs have also been shown to contribute to the migration of OC precursors by producing CXCR4. What is more, MM-EVs promote the bone resorption by augmenting the expression of important osteoclastogenic enzymes (TRAP, CTSK, MMP9). AREG—amphiregulin, BMPs—bone morphogenic proteins, CTSK—Cathepsin K, CXCR4—CXC chemokine receptor type 4, DKK-1—Dickkopf-related protein 1, EGFR—epithelial growth factor receptor, IL—interleukin, lncRunx2-AS1—long-non-coding RNAs Runx2-AS1, miR-135b—micro-RNA 135b, MSCs—mesenchymal stromal cells, MM-EVs—multiple myeloma-derived extracellular vesicles, MMP9—metalloproteinase 9, OBs—osteoblasts, OCs—osteoclasts, OPG—osteoprotegerin, RANKL—receptor activator for nuclear factor κ B ligand, Runx2—Runt-related transcription factor 2, SMAD5—SMAD family member 5, TRAP—thrombospondin-related anonymous protein.

**Figure 4 cancers-14-05575-f004:**
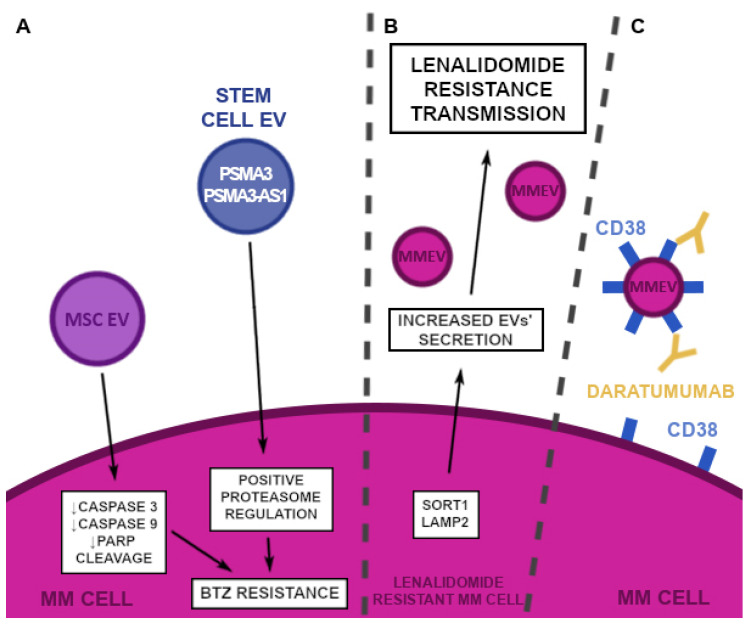
EVs’ involvement in the induction of drug resistance. (**A**) Bortezomib resistance is mediated by EVs in two ways. MM cells, stimulated by BMSC EVs show a decreased caspase 3, 9, and PARP cleavage, therefore the BTZ induction of the apoptosis is blocked. Moreover, stem cell-derived EVs, which contain PSMA3 and PSMA3-AS1, mediate the positive proteasome regulation, thus the apoptosis, due to an intercellular protein buildup of the damaged proteins is inhibited. (**B**) Increased expression of SORT1 and LAMP2 genes in lenalidomide-resistant MM cells lead to the increased release of EVs, which when up-taken by the lenalidomide-sensitive MM cell, confer the resistance. (**C**) EVs of MM cells express CD38 on their surface, therefore it appears that monoclonal antibodies may bind to EVs instead of MM cells, decreasing the bioavailability of the drug. BMSC—bone marrow stromal cell, BTZ—bortezomib, CD—cluster of differentiation, EV—extracellular vesicle, LAMP2—lysosomal associated membrane protein 2, MM—multiple myeloma, MMEV—multiple myeloma extracellular vesicle, PARP—poly-ADP ribose polymerase, PSMA3—proteasome subunit alpha type-3, PSMA3-AS1—proteasome subunit alpha type-3 antisense RNA 1, SORT1—sortilin1.

**Table 1 cancers-14-05575-t001:** Main types of extracellular vesicles and their different characteristics.

Name	Classical Exosomes	Non-Classical Exosomes	Classical Microvesicles	Large Oncosomes	ARMM	Apoptotic Exosomes	Apoptotic Microvesicles	Apoptotic Bodies	Ref.
Category	Exosomes		Microvesicles			Apoptotic Extracellular Vesicles			[17,18,19,20,21,22]
Size	40–150 nm		150–1000 nm	1–10 μm	40–100 nm	<150 nm	100–1000 nm	1–5 μm	
EV class	Small EV	Small EV	Large EV	Large EV	Small EV	Small EV	Small to Large EV	Large EV	
Biogenesis	Exocytosis of MVBs		Direct budding from PM			Apoptosis			
			dependent on RHO GTPases		dependent on ARRDC1 and ESCRT	caspase 3-dependent formation of MVBs and its exocytosis	possibly via direct budding	apoptotic membrane blebbing	
Markers	CD63, CD81, CD9	(CD63-, CD81- and CD9-negative) ^a^	Annexin A1, annexin A2	Annexin A1, annexin A2	ARRDC1	Annexin V			
						CD63, LAMP1, HSP70, S1PR1 and 3			

ARMM—ARRDC1-mediated microvesicle; ARRDC1—arrestin-domain-containing protein 1; CD—cluster of differentiation; ESCRT—endosomal sorting complex required for transport; EV—extracellular vesicle; HSP70—heat shock protein 70; LAMP1—lysosomal-associated membrane protein 1; MVB—multivesicular body; PM—plasma membrane; RHO GTPase—Ras homologous GTPase; S1PR—sphingosine-1-phosphate receptor. ^a^ Hypothetical CD63-, CD81-, CD9-negative exosome.

## Data Availability

Not applicable.

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
