# Peer review of "Extracellular Vesicles in Multiple Myeloma—Cracking the Code to a Better Understanding of the Disease"

_cancers, 2022, doi:10.3390/cancers14225575_

Round 1
Reviewer 1 Report (Previous Reviewer 3)
Thank you for addressing the Reviewer's comments.
Reviewer 2 Report (New Reviewer)
I went through the review response and the entire review I think the authors have answered all the criticisms and the review is now ready to be published
This manuscript is a resubmission of an earlier submission. The following is a list of the peer review reports and author responses from that submission.
Round 1
Reviewer 1 Report
The title does not correspond to the text of the paper.
Everything that has been published about angiogenesis, tumor growth, microenvironment, osteolysis and other topics are chaotically described.
Reviewer 2 Report
The authors provided a well-documented overview of Ev's role in the context of Multiple Myeloma. In addition to this, the review could represent a really interesting point of view in a field so dynamic and rich in potential future applications. If the article is well written, the introduction section could be improved by adding some recent works could be improved with a more general point of view about the application of EVs research in other fields of research adding some recent works related to the importance of exosomes in other diseases (PMID: 34839044). A point that should be fixed is related to the size of different EV: exosomes are 30-150 nm (Table 1). Figure 1 should be increased in quality and definition (objects are too small). The conclusions could be improved and enriched by a discussion related to the need for new technologies for the association of a specific marker with an exosome subtype and the exosome subtype to a particular function and/or group of functions (PMID: 35141731 and others)
I hope that my comments could be useful and I look forward to reading the revised version of the paper.
Good luck.
Reviewer 3 Report
The authors of the present review have focused on the emerging role of extracellular vesicles (EVs) in the pathophysiology of Multiple Myeloma (MM), a deadly hematological malignancy.
Comments:
1. The authors may explain the reason for not including the latest articles in their evaluation e.g. PMID: 34235082; PMID: PMID: 34506129; PMID: 30386953; PMID: 30782029 etc.
2. The article's writing may be improved as it lacks flow as we move from one section to the other.
3. The figure may be enlarged and made more readable and visually appealing to the readers.
Thank you